# Impact of social class on health: The mediating role of health self-management

**Xiaoyong Hu**[1]*, **Tiantian Wang**[1], **Duan Huang**[2], **Yanli Wang**[1], **Qiong Li**[2]

1 Faculty of Psychology, Key Laboratory of Cognition and Personality (Ministry of Education), Southwest University, Chongqing, People's Republic of China, 2 School of Health, Wuhan Sports University, Wuhan, People's Republic of China

* huxiaoyong@swu.edu.cn

## Abstract

### Background

Studies have explored the relationship between social class and health for decades. However, the underlying mechanism between the two remains not fully understood. This study aimed to explore whether health self-management had a mediating role between social class and health under the framework of Socio-cultural Self Model.

### Methods

663 adults, randomly sampled from six communities in Southwest China, completed the survey for this study. Social class was assessed using individuals' income, education, occupation. Health self-management was assessed through evaluation of the health self-management behavior, health self-management cognition, health self-management environment. Physical health and mental health were measured by the Chinese version of Short-Form (36-item) Health Survey, which contains Physical Functioning, Role-Physical, Role-Emotional, Vitality, Mental Health, Social Function, Bodily Pain and General Health. Pearson's correlation was used to examine the associations between major variables. Mediation analyses were performed to explore the mediating role of health self-management.

### Results

Social class positively predicted self-rated health. The lower the social class, the lower the self-reported physical and mental health. Health self-management partially mediated the relationship between social class and self-rated health. That is, the health self-management ability of the lower class, such as access to healthy and nutritious food and evaluate their own health status, is worse than that of the higher class, which leads to physical and mental health inequality between the high and the low classes.

### Conclusion

Health self-management mediated the relationship between social class and health. Promoting health self-management abilities are conducive to improving both physical and mental health.

**Data Availability Statement:** All relevant data are within the manuscript and its Supporting Information files.

**Funding:** This research has been financially supported by Scientific Research Foundation of

Southwest University, grant number SWU2009206. http://swu.edu.cn/ The funders had no role in study design, data collection and analysis, decision to publish, or preparation of the manuscript.

**Competing interests:** The authors have declared that no competing interests exist.

## Introduction

Resident health not only reflects the individual's physical and psychological adaptability, but also reflects the comprehensive strength of a country or region. In recent decades, people's health status has seen a worldwide improvement in correlation with the economic and social development of the country. However, from the perspective of different social groups, the economic development only highlights the class gap in health [1–3]. That is, economic and social development improves the overall health of the whole society, but widens the health gap between different demographics. Therefore, an increasing number of researchers realize that only when social determinants that lead to health differences are fully taken into account, can health intervention that reduce disease and save lives be effective [2, 3].

Social class refers to groups in different positions within the social hierarchy, which was formed by economic and political reasons, among others; there are objective differences in social resources (income, education, and occupation) and subjective differences in perceived social status among these groups [4]. Many studies demonstrate that social classes can positively predict individual health [5–11]. Compared with the lower class, the upper classes have longer life spans, a better health status, and less possibility of suffering from a physical disability [12, 13]. For example, data from the United Kingdom, France, Switzerland, Portugal, Italy, the United States and Australia, published in *Lancet*, shows that lower class is associated with shorter life expectancy [14]. Compared with wealthier people, those in the lower class are approximately 1.5 times more likely to die before the age of 85: among those with a lower socioeconomic status, 55,600 (15.2% males and 9.4% females) died before the age of 85. This study also estimates that lower socioeconomic status can shorten life expectancy by 2.1 years (41% males and 27% females), and indicates that social class, like traditional risk factors such as hypertension, obesity, high alcohol consumption, and a sedentary lifestyle, should be a key factor of health risk [14]. In addition, compared with the upper classes, the lower class experience less happiness [1, 15, 16]; more negative emotions and stress [17, 18]; and more psychological symptoms [19, 20]. For example, research found that social class is closely related to depressive symptoms [21]; that social class is an important factor affecting anxiety; and that lower class children have more anxiety and that their anxiety is often related to psychopathology [22, 23].

Why is it that the lower the social class, the worse the health status? The Socio-cultural Self Model points out that the self is the core mechanism that leads to differences in the health between higher and lower classes [24]. The Socio-cultural Self Model focuses on the role of the self, and regards the self as the product of interaction between individual characters and environmental factors within a specific social and cultural background. The model notes that individual characteristics and environmental factors indirectly influence behavior through the self, which is shaped by social, cultural and individual factors [24]. Under the theoretical framework of the Socio-cultural Self Model, researchers suggested that self-management abilities, such as health self-management, is an important factor explaining class inequalities in health [25–27]. Health self-management is the ability of an individual to take control of their own health [28]. Specifically, it is the ability of an individual to analyze and evaluate her or his own health status and influencing factors in life, and make lifestyle changes, such as be able to actively seek health consultation and guidance, and take intervention measures such as medical treatment, appropriate exercise, or reasonable nutrition for health risk factors, and maintain his or her own health [28–31].

Several studies have confirmed that there are significant differences in the health self-management capabilities among different classes [32, 33]. For example, one research team conducted multi-factor logistic regression analysis on the health self-management of 1028

participants in Hangzhou and found that social class significantly predicted health self-management. Specifically, those with a higher education background have higher health awareness and higher level of health self-management [33]. In addition, a further study shows that health self-management is positively related with individual health status [34, 35].

Actually, many researchers have explored the mediators between social class and health from different aspects, including social (the socioeconomic position of an individual in adulthood [36], adult socioeconomic status [37]), psychological (perceived discrimination [38], social relationships [39], social support [37]), and physical (metabolic alterations [40], cortisol slope [41], subjective social status [13], negative emotions [17]) factors. However, to the best of our knowledge, no studies have yet to directly examine the mediating role of self between social class and health. According to the Socio-cultural Self Model, self is the core mechanism of social class affecting health, and social and individual factors can only work through self. Therefore, in this study, we aimed to test the mediating role of self between social class and health. Under the framework of Socio-Cultural-Self Model, we hypothesized that social class was positively associated with physical and mental health, and that this relationship was mediated by health self-management.

## Material and methods

### Sample and procedure

After obtaining the ethical approval from the Research Ethics Committee of Southwest University (IRB NO. H19070), we collected data from Chongqing, a provincial municipality directly under the central government in Southwest China. Consistent with the purpose to investigate social class and health disparities, we sought a sample that is socially and economically diverse. Sample selection and recruitment began with the type of housing (villa, apartment, low-rent housing). In Yuzhong District and Rongchang District of Chongqing City, one affluent community(villa), one middle class communities(apartment), and one lower class communities (low-rent housing) for each district were randomly selected, and then 125 subjects were randomly selected in each community for questionnaire survey. During the investigation, a total of six well-trained investigators (psychology graduate students) whom divided into three groups, with the support of the Community Neighborhood Committee, investigated six communities one by one through the household survey. Before the survey was distributed, one of the authors explained the purpose of the study to the subjects. It was guaranteed that all of the responses would be kept confidential. All participants provided written informed consent. The survey involved 98 questions and took 20–30 minutes to complete. Totally, surveys were distributed to 750 adults of whom 663 responded to the full survey, resulting in a response rate of 88.4%. The remaining 87 subjects either refused to accept the survey, or more than 50% of the survey questions were not answered. Of these subjects, 41 live in low-rent houses, 28 live in buildings, and 17 live in villas. Among the participants, 55.2% of them were female, 44.8% were male. The average age of the sample was 33.73 years ($SD = 12.61$), and ranged from 18 to 65 years old.

### Measures

**Social class.** Social class was measured by objective socio-economic status (or objective SES) indicators, which involves an intersection of different factors, including income, education, and occupation [42, 43]. To accurately model the interaction of these factors, psychologists have developed many methods to estimate social class, such as factor analysis method, regression equation method, and weighted mean method. Among them, the factor analysis method is widely used [44]. Therefore, we performed factor analysis via structural equation

modeling, with participants' reported income, education, and occupation. Specifically, the factor analysis method includes the following steps:

Firstly, participations rate their education, occupation, and per capita annual household income. According to the standards used by OCED-2012, participations rated their education on the following scale:1 = Never been to school, 2 = elementary school, 3 = junior high school, 4 = secondary technical school, 5 = general high school, 6 = vocational training after high school, 7 = college, 8 = undergraduate, 9 = postgraduate. According to the Chinese Professional Reputation Index [45], participants rated their vocation on the 24 different occupations scale. For example, 1 = senior leading cadres of the government, 21 = ordinary farmers and fishermen, 24 = engaged in such jobs as nanny, hourly worker, manual tricycle driver, etc. Participants rated their per capita annual household income from the following scale: 1 = Less than 3000 RMB, 2 = RMB 3000–5000, 3 = RMB 5000–12000, 4 = RMB 12000–20000, 5 = RMB 20000–30000, 6 = RMB 30000–55000, 7 = Over 55000 RMB.

The second step is to deal with the missing values in each variable by replacing with sequence mean. The third step is to transform the three variables of education, occupation, and per capita annual household income into standard scores. Then, principal component analysis is carried out and the social class variables are calculated according to the following formula: social class = $(\beta 1 \times Z\ education + \beta 2 \times Z\ occupation + \beta 3 \times Z\ income) / \varepsilon f$, in which $\beta 1$, $\beta 2$ and $\beta 3$ are factor loads and $\varepsilon f$ is the characteristic root of the first factor.

**Health self-management.** The health self-management ability scale was developed by Zhao and Huang [46], and includes health self-management behavior (14 items), health self-management cognition (14 items), health self-management environment (10 items) scale, for a total of 38 items. Using the Likert-5 scoring method, each item is scored from 1 to 5. The sum of the three dimensions is the total score of health self-management ability. The higher the total score, the better the health self-management ability. In this study, the internal consistency coefficient (Cronbach's alpha) is 0.94.

**Short-form (36-item) health survey.** Physical and mental health status was measured by the Short-Form (36-item) Health Survey (SF-36). SF-36 has excellent psychological measurement characteristics, and is widely used in the measurement of the general population health status [47]. The Chinese version of SF-36 has sufficient reliability and validity, and its psychological characteristics are similar to the test results of American population samples [48].

The SF-36 consists of 36 items and 8 sub-dimensions, and the applicable population is adults over 14 years old. Excluding one item in the questionnaire that measures health transition, the remaining 35 items belong to eight dimensions related to health: Physical Functioning (PF, 10 items), Role-Physical (RP, 4 items), Role-Emotional (RE, 3 items), Vitality (VT, 4 items), Mental Health (MH, 5 items), Social Function (SF, 2 items), Bodily Pain (BP, 2 items), General Health (GH, 5 items). Each dimension score is the weighted sums of the questions in the section and is directly transformed into a 0–100 scale on the basic assumption that each question carries equal weight. These eight dimensions can be aggregated into two independent comprehensive dimensions: Physical Component Summary (PCS, $\alpha = 0.87$) and Mental Component Summary (MCS, $\alpha = 0.69$). Physical Component Summary consists of PF, RP, BP, and GH and Mental Component Summary consists of RE, VT, MH, SF. The higher the score, the higher the degree of people's mental and physical health status.

**Sociodemographic factors.** Several sociodemographic factors were taken into account, including gender (0 = male, 1 = female) and age.

**Statistical analyses.** SPSS 20.0 was used for data analyses. Firstly, we investigated the general tendency among social class, health self-management, and mental and physical health. We calculated the mean and standard deviation for each variable and the Pearson correlation coefficients between the variables.

Next, we tested the hypothesized mediational model with social class as a predictor, health self-management as a mediator, and general health as an outcome variable, using the AMOS 23.0 statistical package. Concretely, we examined the mediation mechanism of social class on mental and physical health. For the path coefficients a maximum likelihood estimation method was used, and 95% bias-corrected confidence intervals were calculated for all effects using 1,000 bootstrap samples.

## Results

### The sample characteristics

The sample characteristics are shown in **Table 1**. Junior high school accounts for the largest proportion of subjects' education level (29.9%). As for occupation, a large proportion of subjects were ordinary farmers and fishermen (33.5%). In terms of income, 52.5% of the subjects had annual household income per capita less than 5000 RMB.

### Testing for common method variance

Due to the use of self-reported data, Harman's single-factor test was used to rate common method bias. Exploratory factor analysis revealed that the first factor accounted for 17.77% of the total variance and did not explain most of the variance (<40%). Results showed that there was no common method bias in this study.

### Descriptive statistics and correlation analysis

As indicated in **Table 2**, the results demonstrate that social class was positively associated with health self-management ($r = 0.42$, $p<0.01$), mental health ($r = 0.24$, $p<0.01$), and physical health ($r = 0.15$, $p<0.01$). Health self-management was positively related with mental health ($r = 0.22$, $p<001$), and physical health ($r = 0.18$, $p<0.01$).

### Mediating analysis

We used structural equation modeling to test the hypothesis that the association between social class and mental and physical health is mediated by health self-management. Fig 1 shows the results from SEM testing of the hypothesized mediation model. The relationships of age also were considered in this model (for simplicity in presentation, the paths from age to other variables are not shown in Fig 1). The adjustment for age was necessary, since physical health($r = -0.14$, $p<0.01$) and health self-management ($r = -0.18$, $p<0.01$) worsens with age and social class also negatively related with age($r = -0.42$, $p<0.01$).

As shown in Fig 1, using the maximum likelihood estimation, the model fit the data adequately, $\chi 2 = 1.958$, $df = 2$, $\chi 2/ df = 0.979$, Comparative Fit Index (CFI) = 1.000, Normed fit index (NFI) = 0.995, Tucker Lewis index (TLI) = 1.001, Root Mean Square Error of Approximation(RMSEA) = 0.000.

The mediations were observed by evaluating the significance of the indirect paths [49]. We achieved 95% bootstrap confidence intervals (CI) for hypothesized indirect effects using 1,000 bootstrap samples. This is because the bootstrap procedure does not require the normality assumption, which is recommended for testing the mediation effect [50]. Table 3 shows the bootstrap CIs and estimates for each hypothesized indirect effect. It also shows that the CIs for all hypothesized paths did not contain zero, indicating that all of the hypothesized indirect effects were significant. Fig 1 shows that the estimate of the total effect was disaggregated into two indirect effects, which represent the effects mediated by health self-management (Social Class → Health Self-management → Mental Health; Social Class →

**Table 1. Demographic and socioeconomic data.**

| Characteristics | n(%) |
|---|---|
| Female | 366(55.2) |
| Age(years) | |
| 18–25 | 214(32.3) |
| 26–35 | 186(28) |
| 36–45 | 138(20.8) |
| 46–55 | 88(13.3) |
| 56–65 | 37(5.6) |
| Education | |
| 1 = Never been to school | 53(8) |
| 2 = Elementary school | 113(17) |
| 3 = Junior high school | 198(29.9) |
| 4 = Secondary technical school | 31(4.7) |
| 5 = General high school | 128(19.3) |
| 6 = Vocational training after high school | 43(6.4) |
| 7 = College | 67(10.1) |
| 8 = Undergraduate | 27(4.1) |
| 9 = Postgraduate | 3(0.5) |
| Occupational status | |
| 1 = Senior leading cadres of party and government (cadres at ministerial level or above) | 0(0) |
| 2 = Senior professional and technical personnel, such as university professors, well-known scientists and so on | 1(0.2) |
| 3 = Party and government middle-level leading cadres | 5(0.8) |
| 4 = Leaders of government-affiliated institutions | 10(1.5) |
| 5 = Ordinary cadres of party and government organs and institutions | 12(1.8) |
| 6 = Professional and technical personnel in the fields of media, justice and education | 9(1.4) |
| 7 = Director, manager, and middle management of an enterprise | 7(1.1) |
| 8 = Law enforcement officers from taxation and other departments of the Public Security Bureau | 11(1.7) |
| 9 = Ordinary civil servants in party and government agencies and public institutions | 5(0.8) |
| 10 = Medical, engineering, economic and senior professional and technical personnel | 18(2. 7) |
| 11 = Private entrepreneur | 23(3.5) |
| 12 = Factory directors, managers of collective enterprises and middle-level managers of secondary industry enterprises | 17(2.6) |
| 13 = Middle and low-level professional technical personnel | 35(5.3) |
| 14 = Party and government organs and institutions logistics, political work, secretary, financial personnel, etc | 19(2.9) |
| 15 = All kinds of enterprise logistics, political, administrative personnel, salesman, distribution personnel, etc | 38(5.7) |
| 16 = Rural professionals, such as veterinarians, village doctors, etc | 47(7.1) |
| 17 = Small shopkeepers, owners of small workshops and other self-employed persons | 42(6.3) |
| 18 = General staff in business services | 11(1.7) |
| 19 = Industrial workers, production workers in manufacturing, including skilled workers and unskilled workers, etc. | 58(8.7) |
| 20 = professional farmer | 14(2.1) |
| 21 = Ordinary farmer and fishermen | 222(33.5) |
| 22 = Individual laborer | 27(4.1) |
| 23 = Heavy manual workers, such as porters, stevedores, miners, builders, etc | 24(3.6) |
| 24 = Engaged in nanny and part time worker, such as tricycle driver | 8(1.2) |
| Annual household income per capita | |

*(Continued)*

**Table 1.** (Continued)

| Characteristics | n(%) |
|---|---|
| 1 = Less than 3000 RMB, | 135(20.4) |
| 2 = RMB 3000–5000, | 213(32.1) |
| 3 = RMB 5000–12000, | 100(15.1) |
| 4 = RMB 12000–20000, | 116(17.5) |
| 5 = RMB 20000–30000, | 73(11) |
| 6 = RMB 30000–55000, | 26(3.9) |
| 7 = Over 55000 RMB. | 0(0) |

Health Self-management → Physical Health), and the remaining direct effect of social class on mental health that is independent of the mediator. The two indirect paths linking social class and mental (indirect path coefficient 0.061, $p<0.001$) and physical health (indirect path coefficient 0.057, $p<0.001$) were statistically significant. These results suggest that the association between social class and health can be explained through self-health management, which supports our hypothesis.

## Discussion

In this study, we investigated the social class inequality in health and its mechanism. To our knowledge, this is the first study that directly tested the mediation effect of health self-management developed through the Socio-cultural Self Model. Health is not only a research field of medicine and biology, but also one of social science: only when the social determinants that lead to health differences are fully taken into account, can health intervention that reduces disease and save lives be effective. Under this research orientation, this study explored the effect of social class, an important social determinant, on health, and their underlying mechanisms.

One of the main findings of this study is that social class positively predicted self-rated health. Health is the cornerstone of economic and social development, and resident health has both micro and macro levels of significance. The micro level reflects the individual's physical and psychological adaptability; the macro reflects the comprehensive strength of a country or region. According to the World Health Statistics Report 2019, the overall health status of Chinese residents is increasing year by year correlating to the development of economy, medical treatment and education. However, the rise of China's overall health status does not mean that health can develop in a balanced manner among different groups. Empirical studies in major developed countries in Europe have also shown that although economic development has generally improved the health status of the population, it did not reduce the health inequality

**Table 2. The Means and Correlations among central study variables (N = 663).**

| | Mean | SD | 1 | 2 | 3 | 4 | 5 | 6 |
|---|---|---|---|---|---|---|---|---|
| 1. Gender | 0.56 | 0.5 | 1 | | | | | |
| 2. Age | 33.73 | 12.61 | -0.10* | 1 | | | | |
| 3. Social class | 0.05 | 0.78 | 0.06 | -0.42** | 1 | | | |
| 4. Health self-management | 145.27 | 16.73 | 0.04 | -0.18** | 0.42** | 1 | | |
| 5. Mental health | 61.4 | 10.37 | -0.03 | -0.06 | 0.24** | 0.22** | 1 | |
| 6. Physical health | 58.37 | 9.52 | 0.02 | -0.14** | 0.15** | 0.18** | 0.30** | 1 |

Notes: *$p < 0.05$.

** $p < 0.01$(two-tailed test).

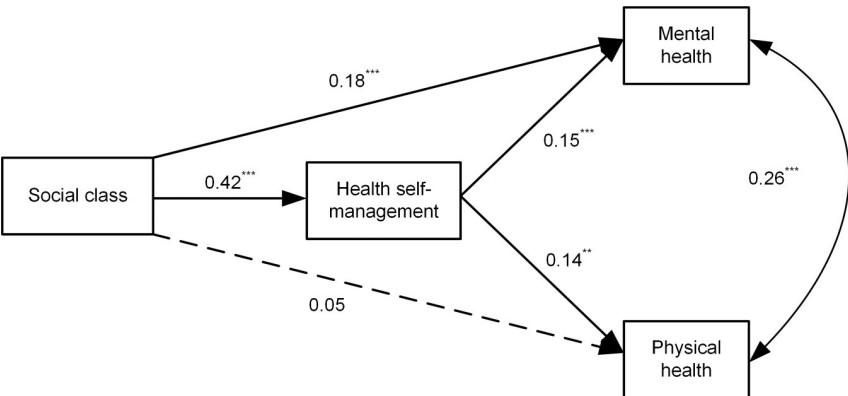

**Fig 1. Empirical mediation model of social class and physical and mental health through health self-management.**
Age are controlled for but are not illustrated for simplicity. **p < .01, ***p < .001.

among the population [12]. The upper class of a population would receive more "health wel-fare" from the social and economic development, thereby increasing the health gradient among different classes [5, 6]. Our results support this viewpoint and confirmed that the lower the social class, the lower the health status. The health inequality is increasing with the deepen-ing of educational, vocational, and income inequalities.

**Table 3. Test of mediation effects of health self-management on the relationship of social class to physical and mental health: Bootstrap results.**

|  | Parameter estimates |
|---|---|
| **Mediation Model, standardized coefficients(95%CI)** |  |
| *Total Effect* |  |
| Social Class → Mental Health | 0.241 (0.175; 0.307)** |
| Social Class → Physical Health | 0.107 (0.031; 0.186)** |
| *Direct effect* |  |
| Age → Physical Health | -0.105 (-0.192; -0.008)* |
| Social Class → Mental Health | 0.180 (0.099; 0.252)** |
| Social Class → Physical Health | 0.050 (-0.042; 0.134) |
| Social Class → Health Self-management | 0.416 (0.355; 0.474)** |
| Health Self-management → Mental Health | 0.147 (0.082; 0.218)** |
| Health Self-management → Physical Health | 0.137 (0.063; 0.201)** |
| *Indirect effect* |  |
| Social Class → Health Self-management → Mental Health | 0.061 (0.034; 0.094)** |
| Social Class → Health Self-management → Physical Health | 0.057 (0.027; 0.086)** |
| **Proportion of the effect of Social Class on Mental Health** |  |
| Mediated by Health Self-management | 25.00% |
| Direct effect | 75.00% |
| **Proportion of the effect of Social Class on Physical Health** |  |
| Mediated by Health Self-management | 52.83% |
| Direct effect | 47.17% |

Bias corrected and accelerated 95% CI, bootstrap resamples = 1,000.

The significance levels are for the standardized solution (*p < 0.05,

** p < 0.01).

Another finding in this study is that health self-management partially mediated the relationship between social class and self-rated health. That is, the lower class has a lower ability of health self-management, which leads to a worse health status. There are many studies have researched the mediators between social class and health, such as the socioeconomic position of an individual in adulthood [36], perceived discrimination [38], negative emotions [17], and health literacy [51, 52]. However, to our knowledge, this is the first study in the field of health to confirm the hypothesis of Socio-cultural Self Model through empirical data. The Socio-cultural Self Model is currently the most systematic elaboration of social class inequality in the health [24–27]:it overcomes the limitations of individual models and structural models, and emphasizes the role of the self which shaped by social culture. Compared with other mediators such as health literacy (health knowledge, motivation and competences), health self-management is a dimension of self and a more fundamental mechanism between social class and health [24]. We supported this theory with evidence that lower classes have lower health self-management abilities than the upper class, such as inability to access to healthy and nutritious food and evaluate their own health status, which causes them to have worse health status than the upper class.

Our results have important implications for the development of relevant theories and the formation of policies. Class gap in health is an important social problem faced by all countries in the world, and since the World Health Organization established the "Health Social Determinants Committee", the research of social determinants of health has been of high importance. The experience of various countries has proved that it is not enough to simply emphasize the biological causes that directly cause the disease, because social determinants such as social class have a greater impact on the health of the population than health services [10, 12]. Our findings suggest that promoting the mobility of socioeconomic status and health self-management abilities are conducive to improving both physical and mental health. Therefore, the findings of our study can provide theoretical guidance for the creation of pertinent health policies, and provide a new framework for the practice of closing the class gap in health.

Overall, this study establishes an integrated framework to examine the mediating effect of health self-management on the relationship between social class and health. However, several limitations of this study must be noted. First, health as an outcome variable is self-reported, which may be different from the objective health status. Second, this study is a correlational study and cannot be used for causal inference. Therefore, future studies can use longitudinal studies or laboratory experiments to explore the mediating role of health self-management between social class and objective health indicators.

## Conclusion

Our results reveal that social class positively predicted self-rated physical and mental health, and that health self-management partially mediated the relationship between the two. More specifically, the lower the class, the lower the health self-management ability, which in turn leads to worse mental and physical health statuses. Revealing the importance of health self-management in the influence of social class on mental and physical health.

## Supporting information

**S1 Fig. Empirical mediation model of social class and physical and mental health through health self-management.** Age are controlled for but are not illustrated for simplicity. $^{**}$p < .01, $^{***}$p < .001.
(TIF)

**S1 Table. Demographic and socioeconomic data.**
(XLSX)

**S2 Table. The Means and Correlations among central study variables (*N* = 663).**
(XLSX)

**S3 Table. Test of mediation effects of health self-management on the relationship of social class to physical and mental health: Bootstrap results.**
(XLSX)

**S1 Data.**
(RAR)

## Author Contributions

**Conceptualization:** Xiaoyong Hu, Duan Huang.

**Data curation:** Yanli Wang.

**Methodology:** Xiaoyong Hu, Tiantian Wang, Yanli Wang, Qiong Li.

**Resources:** Duan Huang.

**Writing – original draft:** Tiantian Wang.

**Writing – review & editing:** Xiaoyong Hu, Qiong Li.

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
