## [Decision Letter · Decision Letter 0]

27 Apr 2021

PONE-D-20-33512

Impact of social class on health: The mediating role of health self-management

PLOS ONE

Dear Dr. Hu,

Thank you for submitting your manuscript to PLOS ONE. After careful consideration, we feel that it has merit but does not fully meet PLOS ONE’s publication criteria as it currently stands. Therefore, we invite you to submit a revised version of the manuscript that addresses the points raised during the review process.

We look forward to receiving your revised manuscript.

Kind regards,

Filipe Prazeres, MD, MSc, Ph.D.

Academic Editor

PLOS ONE

Journal Requirements:

3. Please ensure that you include a title page within your main document. We do appreciate that you have a title page document uploaded as a separate file, however, as per our author guidelines (http://journals.plos.org/plosone/s/submission-guidelines#loc-title-page) we do require this to be part of the manuscript file itself and not uploaded separately.

Reviewers' comments:

Reviewer's Responses to Questions

**Comments to the Author**

1. Is the manuscript technically sound, and do the data support the conclusions?

Reviewer #1: Partly

2. Has the statistical analysis been performed appropriately and rigorously? 

Reviewer #1: Yes

3. Have the authors made all data underlying the findings in their manuscript fully available?

Reviewer #1: Yes

4. Is the manuscript presented in an intelligible fashion and written in standard English?

Reviewer #1: Yes

5. Review Comments to the Author

Reviewer #1: The manuscript reports on a well-executed cross-sectional study aimed to investigate the mediating role of health self-management abilities between social class and health.

Comments

1. Material and Methods. Pg 5 ll106-109. Please explain in more details how the survey was administered, the total number of questions included in the survey and the average completion time

2. Material and Methods. Pg 5 ll106-109. Provide a description of the distribution of non-respondents across the type of housing and of non-participation reasons

3. Material and Methods. Pg 6 ll 138-143. Please provide the full version of the health self-management ability scale as a supplementary material.

4. Results. Please provide a description of the study sample at the beginning of the result section and add a table reporting the characteristics of the study sample.

5. Discussion. Pg 12 ll 243-253. Please discuss your findings in the light of other studies that have showed a mediating role exerted by abilities related to health self-management, such as health literacy skills. Suggested reference:

• Lastrucci V, Lorini C, Caini S; Florence Health Literacy Research Group, Bonaccorsi G. Health literacy as a mediator of the relationship between socioeconomic status and health: A cross-sectional study in a population-based sample in Florence. PLoS One. 2019 Dec 23;14(12):e0227007.

• Van der Heide L, Wang J, Droomers M, Spreeuwenberg P, Rademakers J, Uiters E. The relationship between health, education, and health literacy: results from the Dutch Adult Literacy and Life Skills Survey. J Health Commun. 2013;18(S1): 172–184.

6. PLOS authors have the option to publish the peer review history of their article (what does this mean?). If published, this will include your full peer review and any attached files.

Reviewer #1: No

---

## [Author Response · Author response to Decision Letter 0]

16 Jun 2021

(Reviewer) 1. Material and Methods. Pg 5 ll106-109. Please explain in more details how the survey was administered, the total number of questions included in the survey and the average completion time

(Authors) We have explained in more details how the survey was administered in the revised manuscript (Pg 5 ll105-111). Specifically, the revised contents are as following: During the investigation, a total of six well-trained investigators (psychology graduate students) whom divided into three groups, with the support of the Community Neighborhood Committee, investigated six communities one by one through the household survey. Before the survey was distributed, one of the authors explained the purpose of the study to the subjects. It was guaranteed that all of the responses would be kept confidential. All participants provided written informed consent. The survey involved 98 questions and took 20-30 minutes to complete. 

(Reviewer) 2. Material and Methods. Pg 5 ll106-109. Provide a description of the distribution of non-respondents across the type of housing and of non-participation reasons

(Authors) we have provided a description of the distribution of non-respondents across the type of housing and of non-participation reasons in the revised manuscript (Pg 5 ll113-115). Specifically, the revised contents are as following: The remaining 87 subjects either refused to accept the survey, or more than 50% of the survey questions were not answered. Of these subjects, 41 live in low-rent houses, 28 live in buildings, and 17 live in villas.

(Reviewer) 3. Material and Methods. Pg 6 ll 138-143. Please provide the full version of the health self-management ability scale as a supplementary material.

(Authors) As shown below, this is the full version of the health self-management ability scale and we have put it in the supplementary material. 

The following questions are intended to reflect your views or beliefs about health self-management. Choose from the five options below according to how much you agree with this view or how true it is to you. 

1=Strongly disagree, 2=Disagree, 3=Not sure, 4=agree, 5=Strongly agree.

Eat regular meals a day 1——2——3——4——5

Eat lightly 1——2——3——4——5

Scientific allocation of the ratio of breakfast, lunch and dinner 1——2——3——4——5

Focus on the nutritional mix of different foods each day 1——2——3——4——5

Drink at least 1,200 ml clean water a day 1——2——3——4——5

According to your own situation or professional advice to choose sports, exercise intensity and time 1——2——3——4——5

Make full use of places and equipment for exercise and rest 1——2——3——4——5

Do at least 30 minutes physical activity at least 3 times a week 1——2——3——4——5

The exercise is mainly aerobic 1——2——3——4——5

Replenish water or salt properly after exercise 1——2——3——4——5

Pay attention to the expiry date of the label and the instruction when taking the medicine 1——2——3——4——5

Go to the hospital in time when you feel unwell 1——2——3——4——5

Cooperate with the treatment of medical staff actively when seeking medical treatment 1——2——3——4——5

Have the ability to explain your condition clearly to your doctor when you seek medical advice 1——2——3——4——5

Clean and disinfect daily utensils and equipment regularly 1——2——3——4——5

Keep working or living environment cleanly and tidy 1——2——3——4——5

Pay attention to food hygiene and safety 1——2——3——4——5

Create a good sleep environment 1——2——3——4——5

When indoor air is bad, it can be improved in a variety of ways 1——2——3——4——5

Understand or take advantage of the appropriate health insurance 1——2——3——4——5

Follow or use health-related policies 1——2——3——4——5

Take advantage of health services around you 1——2——3——4——5

Seek help and support from others when needed 1——2——3——4——5

Make full use of the health knowledge gained 1——2——3——4——5

Having good interpersonal relationships can promote health 1——2——3——4——5

Having a good living or working environment can promote health 1——2——3——4——5

Family or friends play a supervisory role in managing your health 1——2——3——4——5

Family or friends can set an example for your own health management 1——2——3——4——5

Vaccination is an economical and effective measure to prevent some infectious diseases 1——2——3——4——5

Rational use of health care drugs or equipment 1——2——3——4——5

Obeying medical advice is good to stay healthy 1——2——3——4——5

Diet, exercise and mentality are important factors of affecting health 1——2——3——4——5

Less smoking, more exercise and a balanced diet can reduce the occurrence of cerebrovascular diseases 1——2——3——4——5

Implement treatment plans or recommendations made by medical staff 1——2——3——4——5

Keep away from the influence of other people's bad habits 1——2——3——4——5

Change unhealthy lifestyles and adopt self-care behaviors 1——2——3——4——5

Deal with the health problems properly 1——2——3——4——5

Have the ability to identify activities or things harmful to health 1——2——3——4——5

(Reviewer) 4. Results. Please provide a description of the study sample at the beginning of the result section and add a table reporting the characteristics of the study sample.

(Authors) We have provided the description and a table of the study sample at the beginning of the result section (Pg 8 ll 188-193.). The added parts are shown below.

The sample characteristics are shown in Table 1. Junior high school accounts for the largest proportion of subjects’ education level (29.9%). As for occupation, a large proportion of subjects were ordinary farmers and fishermen (33.5%). In terms of income, 52.5% of the subjects had annual household income per capita less than 5000 RMB.

Table 1. Demographic and socioeconomic data. 

Characteristics n(%)

Female 366(55.2)

Age(years） 

18-25 214(32.3)

26-35 186(28)

36-45 138(20.8)

46-55 88(13.3)

56-65 37(5.6)

Education 

1= Never been to school 53(8)

2=Elementary school 113(17)

3=Junior high school 198(29.9)

4=Secondary technical school 31(4.7)

5=General high school 128(19.3)

6=Vocational training after high school 43(6.4)

7=College 67(10.1)

8=Undergraduate 27(4.1)

9=Postgraduate 3(0.5)

Occupational status 

1=Senior leading cadres of party and government (cadres at ministerial level or above) 0(0)

2=Senior professional and technical personnel, such as university professors, well-known scientists and so on 1(0.2)

3=Party and government middle-level leading cadres 5(0.8)

4=Leaders of government-affiliated institutions 10(1.5)

5=Ordinary cadres of party and government organs and institutions 12(1.8)

6=Professional and technical personnel in the fields of media, justice and education 9(1.4)

7=Director, manager, and middle management of an enterprise 7(1.1)

8=Law enforcement officers from taxation and other departments of the Public Security Bureau 11(1.7)

9=Ordinary civil servants in party and government agencies and public institutions 5(0.8)

10=Medical, engineering, economic and senior professional and technical personnel 18(2.7）

11=Private entrepreneur 23(3.5)

12=Factory directors, managers of collective enterprises and middle-level managers of secondary industry enterprises 17(2.6)

13=Middle and low-level professional technical personnel 35(5.3)

14=Party and government organs and institutions logistics, political work, secretary, financial personnel, etc 19(2.9)

15=All kinds of enterprise logistics, political, administrative personnel, salesman, distribution personnel, etc 38(5.7)

16=Rural professionals, such as veterinarians, village doctors, etc 47(7.1)

17=Small shopkeepers, owners of small workshops and other self-employed persons 42(6.3)

18=General staff in business services 11(1.7)

19=Industrial workers, production workers in manufacturing, including skilled workers and unskilled workers, etc. 58(8.7)

20=professional farmer 14(2.1)

21=Ordinary farmer and fishermen 222(33.5)

22=Individual laborer 27(4.1)

23=Heavy manual workers, such as porters, stevedores, miners, builders, etc 24(3.6)

24=Engaged in nanny and part time worker, such as tricycle driver 8(1.2)

Annual household income per capita 

1= Less than 3000 RMB, 135(20.4)

2= RMB 3000-5000, 213(32.1)

3= RMB 5000-12000, 100(15.1)

4= RMB 12000-20000, 116(17.5)

5= RMB 20000-30000, 73(11)

6= RMB 30000-55000, 26(3.9)

7= Over 55000 RMB. 0(0)

(Reviewer) 5. Discussion. Pg 12 ll 243-253. Please discuss your findings in the light of other studies that have showed a mediating role exerted by abilities related to health self-management, such as health literacy skills. Suggested reference:

• Lastrucci V, Lorini C, Caini S; Florence Health Literacy Research Group, Bonaccorsi G. Health literacy as a mediator of the relationship between socioeconomic status and health: A cross-sectional study in a population-based sample in Florence. PLoS One. 2019 Dec 23;14(12):e0227007.

• Van der Heide L, Wang J, Droomers M, Spreeuwenberg P, Rademakers J, Uiters E. The relationship between health, education, and health literacy: results from the Dutch Adult Literacy and Life Skills Survey. J Health Commun. 2013;18(S1): 172–184.

(Authors) We thank the reviewer for providing these two important references and We have followed this suggestion and discussed our findings in light of other studies (Pg 15 ll 259-263.). The added parts are shown below: There are many studies have researched the mediators between social class and health, such as the socioeconomic position of an individual in adulthood [36], perceived discrimination [38], negative emotions [17], and health literacy [51,52]. However, to our knowledge, this is the first study in the field of health to confirm the hypothesis of Socio-cultural Self Model through empirical data. The Socio-cultural Self Model is currently the most systematic elaboration of social class inequality in the health [24-27]:it overcomes the limitations of individual models and structural models, and emphasizes the role of the self which shaped by social culture, and regards self as the core mechanism of health differences between higher and lower class. Compared with other mediators such as health literacy (health knowledge, motivation and competences), health self-management is a dimension of self and a more fundamental mechanism between social class and health [24].

---

## [Decision Letter · Decision Letter 1]

2 Jul 2021

Impact of social class on health: The mediating role of health self-management

PONE-D-20-33512R1

Dear Dr. Hu,

We’re pleased to inform you that your manuscript has been judged scientifically suitable for publication and will be formally accepted for publication once it meets all outstanding technical requirements.

Kind regards,

Filipe Prazeres, MD, MSc, Ph.D.

Academic Editor

PLOS ONE

Additional Editor Comments (optional):

Reviewers' comments:

Reviewer's Responses to Questions

**Comments to the Author**

1. If the authors have adequately addressed your comments raised in a previous round of review and you feel that this manuscript is now acceptable for publication, you may indicate that here to bypass the “Comments to the Author” section, enter your conflict of interest statement in the “Confidential to Editor” section, and submit your "Accept" recommendation.

Reviewer #1: All comments have been addressed

2. Is the manuscript technically sound, and do the data support the conclusions?

Reviewer #1: Yes

3. Has the statistical analysis been performed appropriately and rigorously? 

Reviewer #1: Yes

4. Have the authors made all data underlying the findings in their manuscript fully available?

Reviewer #1: Yes

5. Is the manuscript presented in an intelligible fashion and written in standard English?

Reviewer #1: Yes

6. Review Comments to the Author

Reviewer #1: The authors have adequately addressed comments raised in a previous round of review, and the manuscript is now acceptable for publication

7. PLOS authors have the option to publish the peer review history of their article (what does this mean?). If published, this will include your full peer review and any attached files.

Reviewer #1: No

---

## [Editor Report · Acceptance letter]

8 Jul 2021

PONE-D-20-33512R1 

Impact of social class on health: The mediating role of health self-management 

Dear Dr. Hu:

I'm pleased to inform you that your manuscript has been deemed suitable for publication in PLOS ONE. Congratulations! Your manuscript is now with our production department. 

Kind regards, 

on behalf of

Prof. Filipe Prazeres 

Academic Editor

PLOS ONE